# INRCT: An End-to-End Framework for Encoding and Generating Implicit Neural Representation

## Abstract

Current diffusion models based on implicit neural representations (INRs) typically adopt a two-stage framework: an encoder is first trained to map signals into a latent INR space, followed by a diffusion model that generates latent codes from noise. This design requires training and maintaining two separate models, introducing compounded reconstruction errors through the latent-to-data mapping and often leading to increased system complexity. In this work, we propose INRCT, a unified and end-to-end training generative framework for modality-agnostic INR modeling. Instead of operating in the latent space, INRCT performs diffusion directly in the data space by training a single INR hyper-network as a denoiser. Given noisy observations at different noise levels, INRCT predicts the INR for the corresponding clean signal, which is then rendered into data space for supervision. Our training objective coherently integrates a generation loss and a reconstruction loss to jointly support INR generation from noise and INR encoding from real signals within a single model. Extensive experiments on multiple benchmark datasets demonstrate that INRCT achieves superior generation and reconstruction performance compared to existing two-stage generative INR methods, while significantly improving the model efficiency and simplifying model design.

## 1    Introduction

Implicit Neural Representations (INRs) have recently emerged as a powerful framework for modeling continuous signals across various domains, such as images Sitzmann et al. (2020b); Liu et al. (2023a), 3D shapes Park et al. (2019); Sitzmann et al. (2019), and videos Chen et al. (2021); Guo et al. (2025). By representing signals as continuous functions parameterized by a neural network whose inputs are the continuous coordinates and outputs are corresponding values at the coordinates, INRs bypass the limitations of discrete grid-based representations and offer a compact, resolution-independent encoding of complex and high-dimensional data. These modality-agnostic properties make INRs a promising tool for both reconstruction and generative modeling tasks, enabling their deployment in diverse applications such as 3D object synthesis Hong et al. (2024), image super-resolution Zhang et al. (2022), flexible-resolution image generation Chen et al. (2024), and neural radiance field (NeRF) Mildenhall et al. (2021).

Among various generative paradigms built upon INRs, diffusion-based models have recently emerged as the dominant approach, owing to their strong capacity in modeling complex data distributions Ho et al. (2020); Song et al. (2021). Leveraging the ability of INRs to parameterize signals from diverse modalities as continuous functions, recent works Dupont et al. (2022a); You et al. (2023); Park et al. (2024) typically encode each signal into a set of INR parameters and then employ a diffusion model to generate these parameters from noise. This approach inherently supports modality-agnostic signal generation, as it decouples signal representation from data modality. Based on this formulation, adaptations have been applied to different domains, achieving promising results in tasks such as high-resolution image generation Chen et al. (2024) and 3D content generation Erkoç et al. (2023).

However, existing INR-based diffusion models Dupont et al. (2022a); Park et al. (2024); You et al. (2023) rely on a two-stage training process, which often results in inefficiencies and suboptimal per-

formance. As illustrated in Figure 1(a), these models first encode signals into a latent INR space, and then use a separate diffusion process, typically based on DDPM Ho et al. (2020), to generate signals from this latent representation. In this setup, the two stages are trained independently, requiring the maintenance and optimization of two separate neural networks. More critically, the encoder and the denoiser are optimized using fundamentally different loss functions, i.e., reconstruction loss for the encoder and denoising loss for the denoiser, and there is no shared learning objective or gradient flow between them. This design not only *increases the overall model complexity* but also *introduces the potential for compounded reconstruction errors*. The latent space, being an approximation of the data space, can never be a perfect match, making it challenging for the diffusion process to generate samples that faithfully align with the data distribution.

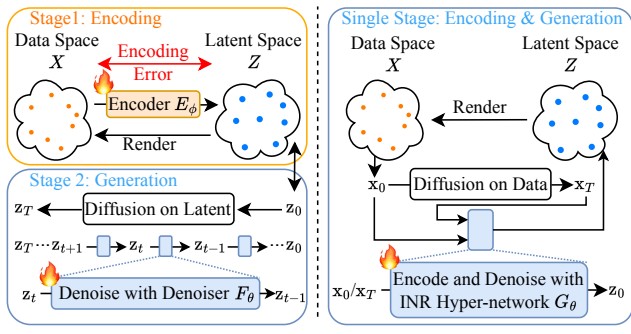

(a) The Existing Two-stage Strategy    (b) Our Unified Single-stage Strategy

Figure 1: (a) Existing INR-based generative models use a two-stage training strategy, where errors in the encoding stage reduce the performance of the generation stage that diffuses on the latent space. (b) Our model supports few-step, and even one-step, denoising, which provides a unified INR encoding and generation framework and enables evaluation in the data space, leading to better generation quality and faster generation process.

To address the above two drawbacks of existing methods, a straightforward solution would be to bypass the latent INR space and perform diffusion directly in the data space, using INR as an intermediate representation that is rendered into the data space and then supervised. However, this approach is *computationally prohibitive* during inference, where DDPM requires hundreds of iterative denoising steps, and rendering the INR at each step introduces substantial overhead. Motivated by recent advances in the few-step diffusion methods, we find that consistency training Song et al. (2023) becomes a rescue. Consistency training Song et al. (2023) learns to map any noisy samples in the diffusion forward process directly to the corresponding clean sample, which enables few-step diffusion. Building on this insight, this paper proposes **INRCT** (Implicit Neural Representation Consistency Training), a novel framework that integrates INR encoding and generation within a single unified model trained in an end-to-end manner, as shown in Figure 1(b). Specifically, INRCT directly maps any noisy data to the INR latent representation of the clean data. The predicted INR latent representation is then differentially rendered into the original data, enabling supervision directly in the data space. We design a cross-modal consistency training objective mapping between the data space and the modality-agnostic INR latent space by coherently integrating a generation loss and a reconstruction loss, enabling INRCT to support both INR generation from noise and INR encoding from real signals within a single model. We further introduce an auxiliary loss term and an efficient training scheduler to improve the convergence.

We summarize the contributions of this work as follows:

- To the best of our knowledge, INRCT is the first end-to-end training framework that unifies INR encoding and INR generation, capable of performing both INR encoding (signal-to-INR) and few-step INR generation (noise-to-INR) within a single model.

- By integrating cross-modal consistency training, INRCT enables end-to-end training, diffusion on data, and few-step INR generation. This not only accelerates signal generation but also eliminates errors associated with two-stage approaches that rely on diffusion in the latent representation, thereby improving generation accuracy.

- The introduction of modality-agnostic INRs into consistency models enables us to extend consistency models to various modalities, allowing few-step diffusion to generate INR for images or 3D NeRF objects.

- Extensive experiments on CIFAR-10, CelebA-HQ, and SRN Cars datasets demonstrate the effectiveness and efficiency of INRCT on INR encoding and generation.

## 2 RELATED WORK

**Implicit neural representations.** By utilizing a neural network (e.g., MLP) to map a coordinate to its corresponding value, INRs have shown great potential in capturing complex continuous functions across various signals, including time-series Fons et al. (2022); Li et al. (2024), images Sitzmann et al. (2020b); Liu et al. (2023a), and 3D scenes Park et al. (2019); Sitzmann et al. (2020a); Liu et al. (2023b); Cardace et al.; Ramirez et al. (2024). Numerous approaches have been proposed to accelerate the process of encoding a given signal into its INR, such as meta-learning Sitzmann et al. (2020a); Tancik et al. (2021); Liu et al. (2023a); Finn et al. (2017) and hyper-networks Chen & Wang (2022); Kim et al. (2023); Zhang et al. (2024); Lee et al. (2023).

**Consistency models.** Diffusion models Sohl-Dickstein et al. (2015); Ho et al. (2020); Song & Ermon (2019) have gained great attention in generative models. These models typically train a generator to reverse the noise corruption in data, thereby estimating the score of the data distribution. They iteratively refine data points sampled from the noise distribution to generate new samples. Many efforts have been made to enhance the inference speed of diffusion models, including faster ODE solvers Song et al. (2021); Lu et al. (2022a;b), predictor-corrector approaches Song et al. (2020) and distillation techniques Salimans & Ho (2022); Meng et al. (2023). Consistency models Song et al. (2023); Song & Dhariwal (2023); Geng et al. (2024); Lu & Song (2024) are a novel class of diffusion models designed for few-step sampling while preserving high-quality generation. They establish a consistency mapping that directly associates any point along the ODE trajectory with its original point, allowing for rapid one-step generation. However, they are only applied to array representations of images and have not shown more application on the more complex signals.

**Generative models based on INRs.** Recent advancements in INR-based generative models have led to various approaches for generating signals across different modalities. Ha (2016) trained GANs and VAEs using functional representations of the MNIST dataset. Works like INR-GAN Skorokhodov et al. (2021) and CIPS Anokhin et al. (2021) use GANs to generate high-quality continuous image functions. Some works focus on shape generation with GAN Kleineberg et al. (2020); Chen & Zhang (2019), VAEs Mescheder et al. (2019), score-based models Cai et al. (2020) and auto decoders Park et al. (2019); Atzmon & Lipman (2020). Additionally, GAN-based methods Schwarz et al. (2020); Niemeyer & Geiger (2021); Chan et al. (2021); DeVries et al. (2021) have been employed to generate NeRF signals, enabling photorealistic 3D-aware image synthesis. Two-stage methods for NeRF generation Karnewar et al. (2023); Erkoç et al. (2023) and high-resolution image synthesis Chen et al. (2024) have also been explored. Modality-agnostic methods like GEM Du et al. (2021) and GASP Dupont et al. (2022b) utilize latent interpolation and GANs to model INR weight distributions, enabling generation across various domains.

The most closely related works to ours are the two-stage modality-agnostic INR generation methods with diffusion models, including Functa Dupont et al. (2022a), DDMI Park et al. (2024), and mNIF You et al. (2023). They first encode the signals into the latent space parameterized by the modulated weights Dupont et al. (2022a), context vectors You et al. (2023), or grid feature Park et al. (2024). Then an extra diffusion model, i.e., DDPM, is applied to the latent space for new sample generation. However, these two-stage approaches lead to error accumulation, with the performance of the generation stage being highly dependent on the reconstruction accuracy of the encoding stage. Additionally, generation with DDPM requires multiple inference steps, which reduces efficiency.

## 3 PRELIMINARIES FOR DIFFUSION MODELS AND CONSISTENCY TRAINING

We firstly present the preliminaries for diffusion models Ho et al. (2020) and consistency training Song et al. (2023) before we present our INRCT. Introduced in consistency models Song et al. (2023), consistency training is a new family of diffusion models that enables training from scratch and supports few-step, even one-step, generation. The core concept underlying consistency models is the probability flow ODE (PF-ODE).

**Diffusion models.** Let $p_{\text{data}}(\mathbf{x}_0)$ denote the data distribution. Diffusion models (DMs) perturb data distributions by progressively adding independent and identically distributed (i.i.d.) Gaussian noise with a time-dependent standard deviation $\sigma(t)$, where $\sigma(0) = \sigma_{\min}$ and $\sigma(T) = \sigma_{\max}$. This noise addition is performed over the interval $t \in [0, T]$ and can be described by a stochastic differential equation (SDE). The reverse-time sampling process corresponds to solving the reverse SDE, and

prior work Song et al. (2020) has shown that this SDE has an equivalent ordinary differential equation (ODE), known as PF-ODE. The PF-ODE shares the same marginal probability densities as the SDE, providing a deterministic way to model the noise perturbation process. Prior works Karras et al. (2022) set $\sigma(t) = t$ and describe the PF-ODE as:

$$\frac{d\mathbf{x}_t}{dt} = -t\nabla_{\mathbf{x}_t}\log p_t(\mathbf{x}_t) = \frac{\mathbf{x}_t - f(\mathbf{x}_t, t)}{t}, \tag{1}$$

where $\nabla_{\mathbf{x}_t}\log p_t(\mathbf{x}_t)$ is the score function, and $f(\mathbf{x}_t, t)$ is a denoising function that predicts the clean image $\mathbf{x}_0$ from the noisy input $\mathbf{x}_t$. We treat time as interchangeable with the noise level. And data $x$ can also be replaced with latent $z$, allowing diffusion directly in the latent space.

**Consistency training.** Consistency models are defined based on the PF-ODE in Eq. 1, and defines a bijective mapping between data distribution and noise distribution. They aim to learn a *consistency function* $f(\mathbf{x}_t, t)$ that maps a noisy image $\mathbf{x}_t$ back to the original clean image $\mathbf{x}_0$, such that:

$$f(\mathbf{x}_t, t) = \mathbf{x}_0. \tag{2}$$

Importantly, the consistency function must adhere to a boundary condition at $t = 0$. Previous works Song et al. (2023) enforce this condition by parameterizing the consistency model as:

$$f_\theta(\mathbf{x}_t, t) = c_{\text{skip}}(t)\mathbf{x}_t + c_{\text{out}}(t)F_\theta(\mathbf{x}_t, t), \tag{3}$$

where $\theta$ represents the model parameters, $F_\theta$ denotes the trainable denoising network, and $c_{\text{skip}}(t)$ and $c_{\text{out}}(t)$ are time-dependent scaling factors satisfying $c_{\text{skip}}(0) = 1$ and $c_{\text{out}}(0) = 0$. This specific parameterization ensures compliance with the boundary condition by design.

Consistency models are trained by minimizing a metric between adjacent points on the sampling trajectory. Specifically, we follow ECT Geng et al. (2024) and consider overlapping intervals under the continuous-time schedule, enabling a probabilistic decomposition $p(t, r) = p(t)p(r|t)$. The time variable $t$ is continuously sampled from a predefined noise distribution $p(t)$, and $r$ is drawn from the distribution $p(r|t)$ with $0 < r < t$. The model is trained to minimize the consistency objective:

$$\arg\min_\theta \mathbb{E}\left[w(t)d\left(f_\theta(\mathbf{x}_t, t), f_{\theta^-}(\widetilde{\mathbf{x}}_r, r)\right)\right], \tag{4}$$

where $w(t)$ is a weighting function that adjusts the importance of different time intervals, $d(\cdot, \cdot)$ is a distance metric (e.g., $L_2$ norm), $f_{\theta^-}$ is an exponential moving average (EMA) of the past values of consistency function $f_\theta$ and $\widetilde{\mathbf{x}}_r = \mathbf{x} + r\mathbf{z}$ can be approximated with the same $\mathbf{x} \sim p_{\text{data}}(\mathbf{x})$ and $\mathbf{z} \sim \mathcal{N}(\mathbf{0}, \boldsymbol{I})$ to calculate $\mathbf{x}_t = \mathbf{x} + t\mathbf{z}$. Therefore, the consistency training objective $\mathcal{L}_{\text{CT}}$ is:

$$\mathcal{L}_{\text{CT}} = \mathbb{E}\left[d\left(f_\theta\left(\mathbf{x} + t\mathbf{z}, t\right), f_{\theta^-}\left(\mathbf{x} + r\mathbf{z}, r\right)\right)\right]. \tag{5}$$

## 4 ENCODING AND GENERATING INRS WITH CONSISTENCY TRAINING

We present a disucssion about the error accumulation of the two-stage training process in Appendix A. To address such a problem, we introduce INRCT, a unified and end-to-end training generative framework for modality-agnostic INR modeling. We first present the overall framework in section 4.1, including how to use a single model to simultaneously support INR encoding and INR generation. Then we present how to train INRCT in an end-to-end manner in section 4.2.

### 4.1 OVERALL FRAMEWORK

Let $I : \mathbf{c} \to \mathbf{s}$ be a continuous signal defined over a bounded domain, where $\mathbf{c} \in \mathbb{R}^C$ denotes the coordinate space and $\mathbf{s} \in \mathbb{R}^S$ the signal value space. For example, an image can be viewed as a function mapping 2D coordinates to RGB values, while a 3D object or scene can be represented as NeRF Mildenhall et al. (2021) that maps 3D spatial coordinates and view directions $v \in \mathbb{R}^3$ to densities and emitted RGB colors. INRs Liu et al. (2023a) approximate the signal $I$ using a coordinate-based neural network $M_\beta$, with parameters $\beta$. Once the neural network satisfies $M_\beta \approx I$, the parameter set $\beta$ serves as a compact implicit representation or latent code of the signal.

We denote the observation set $O$ as a collection of partial measurements from the signal, defined as $O = \{T_i(I)\}_{i=0}^{|O|-1}$, where each transformation $T_i$ depends on the signal modality. In the case of a

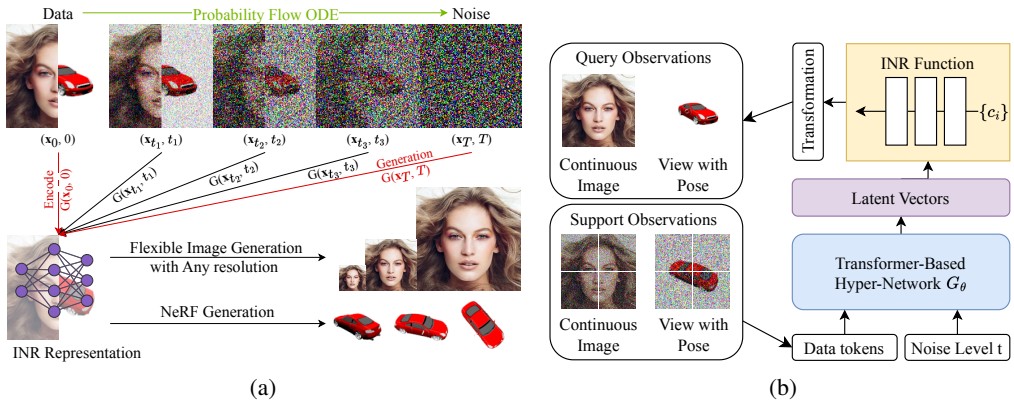

Figure 2: (a) Given a PF-ODE that smoothly converts data to noise, INRCT learn a hyper-network $G$ as the denoiser to map any points on the ODE trajectory to the INR representation of the data. It unifies INR encoding and generation, providing INR encoding as $G(x_0, 0)$ and the INR generation process as $G(x_T, T)$. (b) Data flow of INRCT. It accepts the support observations and the noise level as the input and predicts latent vectors for the INR function for the target signals, which then would be transformed to query observation for supervision.

continuous image, each observation corresponds to the evaluation of $I$ at a 2D coordinate $c_i$, i.e., $T_i(I) = I(c_i)$, which yields the RGB value at pixel $i$. For 3D NeRFs, the observation is acquired via a rendering function $R$, such that $T_i(I) = R(I, r_i)$, where $r_i$ is a camera ray and $R$ integrates radiance along the ray based on the field $I$. This abstraction allows different signal modalities (e.g., images and neural fields) to be handled within a unified mathematical framework.

As presented in Figure 2(a), we inject i.i.d. Gaussian noise with standard deviation $\sigma(t)$ into the observation set $O$ and form a PF-ODE that smoothly converts data into noise. This process yields a noisy observation set $O_t$ corresponding to each time step $t$. The hyper-network $G_\theta$ with learnable parameters $\theta$ is trained to denoise the observations at any noise level $t$, and to predict the INR parameters $\beta$ corresponding to the clean signal:

$$G_\theta(O_t, t) = \beta. \tag{6}$$

Under this formulation, INR encoding corresponds to the special case where $t = 0$, i.e.,

$$\beta_{pred} = G_\theta(O, 0), \tag{7}$$

where the network encodes clean observations into the INR latent code. Similarly, INR generation corresponds to the case where the input is pure noise with the noise level is maximal:

$$\beta_{gen} = G_\theta(Tz, T), \tag{8}$$

where the network acts as a one-step denoiser and generates an INR representation for clean new data from noise. This formulation enables a single model to unify both INR encoding and generation via consistency training over the PF-ODE.

## 4.2 INR-BASED CONSISTENCY TRAINING

In this part, we present how to train a few-step INR denoiser by introducing a cross-modal consistency training objective, where the key idea is to map noisy data into a modality-agnostic INR representation using an INR hyper-network, and then render it back to the clean data for supervision.

As presented in Figure 2(b), to train the hyper-network $G_\theta$, we randomly choose two subsets from the full observation set $O$ and form a support set $O^s$ and a query set $Q^q$, where $O^s$ is the input of the hyper-network and $O^q$ provides the supervised signals to train the hyper-network. For image, $O^s$ and $O^q$ are both the target images. For 3D NeRF, $O^s$ and $O^q$ are projections of the 3D object from views with different poses. We then introduce an i.i.d. Gaussian noise with noise $\sigma(t)$ to both the support observation set and the query observation set to form a noisy support set $O_t^s$ and a noisy query set $O_t^q$. The hyper-network $G_\theta$ is required to denoise $O_t^s$ conditioned on noise level $t$ and predict $\beta$ corresponding to the clean observations:

$$G_\theta(O_t^s, t) = \beta. \tag{9}$$

To evaluate the predicted $\beta$, the transformation from the query observation set, named $T_q$, is used on the predicted $\beta$ to obtain the predicted query observation set:

$$\hat{O}_q = T_q(M_\beta) = T_q(M_{G_\theta(O_t^s, t)}). \tag{10}$$

**Generation objective.** We establish an INR-based consistency function to map a noisy observation set $O_t$ to the clean observation set $O$ with $O_t^q$ as the supervised signal:

$$f(O_t, t) = f(O_t^s, O_t^q, t) = O_q, \tag{11}$$

with the boundary condition defined as:

$$\begin{aligned} f_\theta(O_t^s, O_t^q, t) &= c_{\text{skip}}(t)O_t^q + c_{\text{out}}(t)\hat{O}_q, \\ &= c_{\text{skip}}(t)O_t^q + c_{\text{out}}(t)T_q(M_{G_\theta(O_t^s, t)}). \end{aligned} \tag{12}$$

To learn such a INR-based consistency function, we define a generation objective to enforce the denoising results for any two adjacent noisy points in the PF-ODE to be the same:

$$\mathcal{L}_{\text{gene}} = \mathbb{E}\left[d\left(f_\theta\left(O^s + t\mathbf{z}, O^q + t\mathbf{z}, t\right), f_{\theta^-}\left(O^s + r\mathbf{z}, O^q + r\mathbf{z}, r\right)\right)\right]. \tag{13}$$

We use $\mathcal{L}_{\text{gene}}$ to train the hyper-network $G_\theta$ so that it can denoise a randomly sampled noise from noise level $T$ and generate meaningful INR, achieving new INR generation as defined in Equation 8.

**Reconstruction objective.** Due to the boundary condition, the consistency function at $t = 0$ has $c_{\text{skip}} = 1$ and $c_{\text{out}} = 0$, which prevents the output from being derived from $G_\theta$:

$$f_\theta(O_t, t | t = 0) = 1 \times O_t^q + 0 \times T_q(M_{G_\theta(O_t^s, t | t=0)}). \tag{14}$$

In other words, $\mathcal{L}_{\text{gene}}$ does not provide a supervisory signal for $G_\theta$ at $t = 0$. This approach is well-suited for the original consistency models, as the scenario of inputting $t = 0$ into $G$ does not occur during the generation of new samples. However, we require $G_\theta$ to encode the signal at the noise level $t = 0$. To achieve this, we introduce a reconstruction objective $\mathcal{L}_{\text{recon}}$ to provide a supervisory signal for the hyper-network $G_\theta$ for INR encoding when $t$ is very small ($t < t_{\epsilon_1} = t_{min} + \epsilon$):

$$\mathcal{L}_{\text{recon}} = \mathbb{E}\left[d\left(T_q(M_{G_\theta(O_t^s, t | t < t_{min} + \epsilon)}), O_q\right)\right]. \tag{15}$$

**Diffusion loss at small time-steps.** We notice that for small noise levels $t$, the target of $f_\theta(O_t^s, O_t^q, t)$ can be well approximated by the ground truth $O^q$ instead of approximating by the EMA estimator $f_{\theta^-}(O_r^s, O_r^q, r)$, as the prediction errors may accumulate as integrating $t$ Geng et al. (2024). To address this, we propose a diffusion loss:

$$\mathcal{L}_{\text{diff}} = \mathbb{E}\left[d\left(f_\theta(O_s^t, O_q^t, t < t_{\epsilon_2}), O_q\right)\right], \tag{16}$$

for small noise levels ($t < t_{\epsilon_2}$) to stabilize training. This method avoids applying diffusion loss to large $t$, preventing optimization errors for consistency training. We validate this diffusion loss via experiments in ablation studies.

The total training objective of INRCT is the weighted summation of $\mathcal{L}_{\text{gene}}$, $\mathcal{L}_{\text{recon}}$ and $\mathcal{L}_{\text{diff}}$:

$$\mathcal{L}_{\text{INRCT}} = \lambda_1 \mathcal{L}_{\text{gene}} + \lambda_2 \mathcal{L}_{\text{recon}} + \lambda_3 \mathcal{L}_{\text{diff}}, \tag{17}$$

which provides a unified INR encoding and generation framework and a stable training process.

**An efficient training scheduler.** We observe that directly optimizing the hyper-network $G$ to predict the INR from a noisy signal slows down consistency training in the early stage and negatively impacts final generation performance. This is because, although $\mathcal{L}_{\text{recon}}$ is included, it is only applied during consistency training for cases where $t < t_{\epsilon_1} = t_{\min} + \epsilon$, which occupies a very small portion of the domain of $t$. Consequently, the hyper-network learns the INR encoding process at a slower rate. To address this issue, we introduce an efficient training scheduler to improve the convergence and stability of the training process. Specifically, in the early phase of training, we uniformly sample $t$ with restriction to $0 < t < t_{\epsilon_1}$, and set the weights $\lambda_1$ and $\lambda_3$ to zero so that only the reconstruction loss $L_{\text{recon}}$ is optimized. This enables the hyper-network to quickly develop proficiency in INR encoding. In the ablation study, we show that a hyper-network with a well-initialized INR encoding capability achieves better INR generation performance. We present the training algorithm in Algorithm 1.

Table 1: Table for image generation performance of different generative models on the CIFAR-10 dataset with $32^2$ resolution.

| CIFAR-10-$32^2$ | NFE ↓ | FID ↓ | IS ↑ |
|---|---|---|---|
| Discrete Image Generation Methods | | | |
| PixelIQN | 1 | 49.46 | 5.29 |
| NCSNv2 | - | 31.75 | - |
| StyleGAN2 | 1 | 3.26 | 9.74 |
| DDPM | 1000 | 3.17 | 9.46 |
| DDIM | 10 | 8.23 | - |
| CT | 1 | 8.70 | 8.49 |
| CT | 2 | 5.83 | 8.85 |
| Continuous INR Generation Methods | | | |
| DPF | 1000 | 15.10 | 8.43 |
| GEM | - | 23.83 | 8.36 |
| INRCT (Ours) | 1 | 15.14 | 8.29 |
| **INRCT (Ours)** | **2** | **12.25** | **8.46** |

Table 2: Table for the image generation performance of the INR-based diffusion models and other generative models on the CelebA-HQ dataset with $64^2$ resolution.

| CelebA-HQ-$64^2$ | NFE ↓ | FID ↓ | P ↑ | F1 ↑ |
|---|---|---|---|---|
| Non Diffusion-Based Generative Methods | | | | |
| GEM | - | 30.4 | 0.642 | 0.562 |
| GASP | - | 13.5 | 0.836 | 0.620 |
| CIPS | - | 15.4 | - | - |
| Diffusion-based Generative Methods | | | | |
| DPF | 1000 | 13.2 | 0.866 | 0.634 |
| Functa | 1000 | 40.4 | 0.577 | 0.536 |
| mNIF(S) | 1000 | 21.0 | 0.787 | 0.609 |
| DDMI | 1000 | 9.74 | - | - |
| INRCT(Ours) | 1 | 12.7 | 0.799 | 0.613 |
| **INRCT(Ours)** | **2** | **7.88** | **0.868** | **0.635** |

## 5 EXPERIMENTS

### 5.1 SETTING

**Datasets.** We evaluate our model with the mainstream generative models on the image dataset CIFAR-10 dataset Krizhevsky et al. (2009), and then we follow Functa Dupont et al. (2022a), DDMI Park et al. (2024) and mNIF You et al. (2023) to compare our models with the INR-based generative models on the image datasets CelebA-HQ dataset Karras (2017) for image encoding and generation, and on SRN Cars dataset Sitzmann et al. (2019) for 3D NeRF encoding and generation. Each object in SRN Cars dataset contains several projections from different views, where NeRF will learn its 3D appearance so that it can be projected into any continuous views.

**Implementations.** We configure the hyper-network $G_\theta$ with a Transformer-based architecture. The INR function $M_\beta$ is an attention-based INR function Zhang et al. (2024) for image modality with $\beta$ as the instance-specific representation vector, and is the MLP-based INR function Chen & Wang (2022) for NeRF modality with $\beta$ as the instance-specific weights. More detailed configurations, including the schedule for consistency training, the optimization procedures, and detailed network architectures, are in Appendix B.

**Evaluation metrics.** The quantitative generation performance is evaluated using Frechet Inception Distance (FID) Heusel et al. (2017), Inception Score (IS) Salimans et al. (2016), Precision (P) Kynkäänniemi et al. (2019), and F1 score (F1) Kynkäänniemi et al. (2019). Following Song et al. (2023) and Dhariwal & Nichol (2021), we generate 50,000 images to ensure reliable evaluation metrics. The efficiency of generation is evaluated with the number of function evaluations (NFEs). We use the PSNR to evaluate the INR encoding and reconstruction performance.

### 5.2 RESULTS

**INR generation performance.** For image generation, we present the quantitative comparison of CIFAR-10 dataset in Table 1 and comparison of CelebA-HQ dataset in Table 2. On the CIFAR-10 dataset, with only a two-step evaluation, INRCT achieves better FID and IS compared to all other domain-agnostic INR-based generation models with more steps. We also include the results of the models focusing on discrete image generation on CIFAR-10 dataset in Table 1, and the results show that INRCT substantially closes the performance gap between diffusion models operating on continuous functions (e.g., DPF) and those on discrete grids (e.g., DDPM/CT). We notice that there is a little performance gap between INRCT and CT Song et al. (2023), which is primarily due to the cross-modality transformation in INRCT, where the input and output of the network are different, while the original CT maintains consistent input-output modalities throughout its U-Net, making its optimization simpler and more straightforward. However, the cross-modal transformation is also an advantage of INRCT, which extends CT to different signal modalities.

In addition, we present more comparisons of INRCT with the INR-based generative models on the CelebA-HQ dataset in Table 2. Thanks to the end-to-end training strategy and diffusion on the

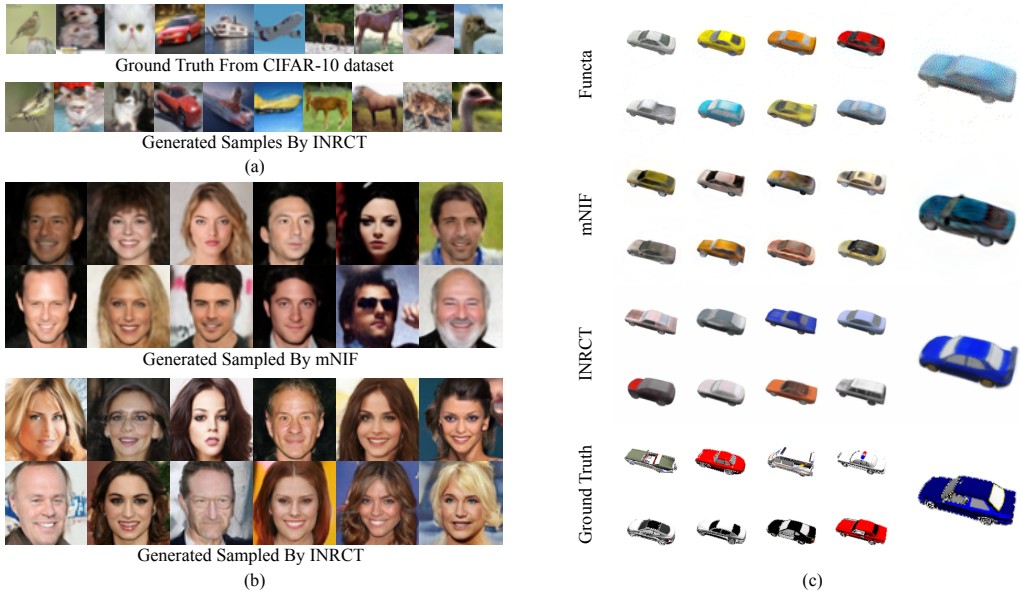

Figure 3: (a) Visual performance of INRCT on the CIFAR-10. We observe that INRCT can generate realistic and diverse images for each class in the CIFAR-10 dataset. (b) Comparison of the visual performance of INRCT and mNIF on the CelebA-HQ. The results from INRCT are more diverse and more realistic. (c) Comparison of the visual performance of INRCT, mNIF, and Functa on the SRN-Cars for NeRF generation. The noise is still remaining in the results from Functa, and the colors in the results from mNIF are chaotic, showing the errors introduced by those two-stage methods.

Table 3: Table for 3D NeRF generation performance comparison on the SRN Cars dataset with $128^2$ resolution.

| SRN Cars-$128^2$ | NFE ↓ | FID ↓ |
|---|---|---|
| Functa | 1000 | 80.3 |
| mNIF | 1000 | 79.5 |
| **INRCT(Ours)** | **1** | **51.3** |

Table 4: Table for signals reconstruction performance on CelebA-HQ dataset with resolution $64^2$ and SRN Cars dataset with resolution $128^2$.

| Reconstruction (PSNR) ↑ | CelebA-HQ | SRN Cars |
|---|---|---|
| Functa | 26.6 | 24.2 |
| mNIF(S) | 31.5 | 25.9 |
| **INRCT(Ours)** | **40.9** | **26.8** |

Table 5: Table for the ablation study on the CelebA-HQ $64^2$ dataset. The effectiveness of each component of the designed cross-modal training objective and the proposed efficient training scheduler (ETS) is discussed. We report the two-step generation FID and one-step reconstruction PSNR to reflect the generation and reconstruction performance separately.

| Setting | | | | FID↓ | PSNR↑ |
|---|---|---|---|---|---|
| $L_{gene}$ | $L_{recon}$ | $L_{diff}$ | ETS | | |
| ✗ | ✓ | ✗ | ✗ | 389 | 49.2 |
| ✓ | ✗ | ✗ | ✗ | 15.93 | 10.7 |
| ✓ | ✓ | ✗ | ✗ | 19.31 | 45.1 |
| ✓ | ✓ | ✓ | ✗ | 9.30 | 37.4 |
| ✓ | ✓ | ✓ | ✓ | **7.88** | **40.9** |

data space, INRCT outperforms those two-stage diffusion methods and non-diffusion-based INR generative models. We also present some visual results of the CIFAR-10 dataset in Figure 3(a) and the CelebA-HQ dataset in Figure 3(b). These visual results prove that INRCT can generate realistic and diverse signals that match the distribution of training samples. More results are in Appendix C.

For 3D NeRF generation, we present the quantitative comparison of the SRN Cars dataset in Table 3. We observe a significant improvement in the 3D NeRF generation quality with INRCT compared to other two-stage methods. As shown in Figure 3(c), Functa's generated images still contain residual noise, while mNIF's generated images have chaotic colors and deviate considerably from the training samples. In contrast, generated 3D NeRF from INRCT images closely match the training samples in both color and appearance, offering better visual quality and diversity. These results further demonstrate the superior accuracy of our end-to-end training strategy for INR generation. More results about the diverse samples and the consistent projective views are presented in the Appendix C.

**INR encoding performance.** We present quantitative results of our method's INR encoding in Table 4, including image INR encoding on CelebA-HQ and 3D NeRF encoding on SRN-Cars. The existing two-stage INR generative models require encoding signals into relatively small represen-

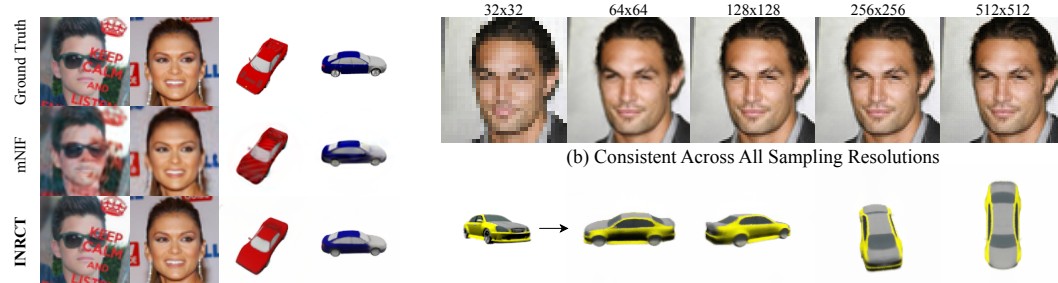

(a) Visual Performance of INR Encoding    (b) Consistent Across All Sampling Resolutions

(c) Novel View Synthesis From a Single View

Figure 4: (a) Comparison of the visual performance of INR encoding for INRCT and mNIF on the CelebA-HQ and SRN-Cars dataset. We find that INRCT achieves better reconstruction performance for INR encoding than mNIF. (b)INRCT can sample any-resolution images without artifacts (c) INRCT can generate 3D contents which contains consistent views from a single view.

tation vectors to enable an efficient diffusion process on the second generation stage, limiting their encoding ability and leading to poor reconstruction accuracy. In contrast, our model combines encoding and generation within a single model. This avoids the restriction on the size of the representation vector. Furthermore, our method benefits from a Transformer-based hyper-network with a large number of parameters, which significantly enhances the model's expressiveness and capacity to encode high-fidelity INR representations. The results in Figure 4(a) also support this conclusion, where the reconstructed results from INRCT are clearer and realistic than mNIF You et al. (2023).

### 5.3 DISCUSSION

**Ablation study.** In Table 5, we present our ablation study. We observe that using the generation objective alone does not yield good reconstruction PSNR, while using the reconstruction objective alone doesn't achieve good generation FID. This result shows that the generation objective is the key to enabling few-step generation, while the reconstruction objective ensures effective INR encoding. Furthermore, by progressively adding the diffusion loss at small time steps and the efficient training scheduler, the few-step generation and encoding performance of INRCT are further improved.

**Flexible image generation.** INRCT inherits the ability of INR and can generate images at any resolution. It can generate a continuous image which is parameterized as a continuous function defined in a bounded range, rather than directly generating discrete samples. With the continuous functions for the images, we can sample at any resolution by specifying specific coordinates within the range and produce artifact-free images at any resolution. We show this capability in Figure 4(b).

**Encoding as single-image-to-3D.** We observe that Functa and mNIF are based on meta-learning or auto-decoder for INR encoding. They need to use all observations as supervision to encode the signal, which requires multiple views. In contrast, our method achieves INR encoding with prediction by a hyper-network. We directly predict the entire NeRF signal from just a few support views (could be as minimum as one support view). This means that our INR encoding for NeRF objects is actually a single-image-to-3D process, enabling novel view synthesis from a single view. We demonstrate the capability of our model in Figure 4(c).

### 6 CONCLUSION

In this paper, we address the error and speed issues in current INR-based diffusion models. They rely on a two-stage process, encoding followed by generation. Due to the need for many iterations in DDPM, it is impractical to perform evaluation in the data space. Therefore, we introduce INRCT, the first end-to-end training generative framework for modality-agnostic INR modeling. We design a cross-modal consistency training objective to enable few-step generation, giving our model the efficiency to perform diffusion directly in the data space. This eliminates the error accumulation from diffusion in the latent space. One limitation of INRCT lies in its scalability. When applied to high-dimensional data, consistency training in the data space demands substantial computational resources, which constrains its applicability to large-scale scenarios. In future work, we plan to integrate modality-specific INR functions to improve efficiency and expressiveness, thereby enhancing encoding and generation quality and enabling SOTA performance on complex signals.

ETHICS STATEMENT

This work does not involve human subjects, personal data, or sensitive information. All datasets used in this study are publicly available and widely adopted in the research community. We have carefully followed the corresponding licenses and terms of use. Our research does not introduce potential harms to individuals, groups, or society, and it does not promote unfairness, discrimination, or security/privacy risks. The methods and results presented are intended solely for academic research purposes. We confirm that the study adheres to the ICLR Code of Ethics and complies with ethical standards of research integrity.

REPRODUCIBILITY STATEMENT

We have made extensive efforts to ensure the reproducibility of our work. The main paper provides a comprehensive description of the proposed methodology, training objectives, and evaluation metrics. Detailed implementation settings, hyperparameters, and training procedures are provided in the appendix. To further facilitate reproducibility, we will release the source code upon acceptance of the paper.

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

## A  FORMAL DISCUSSION FOR THE LIMITATIONS OF THE TWO-STAGE TRAINING PROCESS

In the second stage of the two-stage training framework, a denoising diffusion probabilistic model (DDPM) is trained in the latent space. Let $x$ denote the input data, $E$ denote the encoder, and $R$ denote the renderer (e.g., an implicit neural representation decoder), such that the latent variable is $z_0 = E(x)$ and the reconstructed data is $\hat{x} = R(z_0)$. The encoder $E$ is trained in the first encoding stage using a reconstruction loss:

$$\mathcal{L}_{\text{recon}} = \|R(E(x)) - x\|^2. \tag{18}$$

In the second stage, the diffusion model $F_\theta$ learns to denoise latent variables via the standard noise prediction objective:

$$\mathcal{L}_{\text{denoise}} = \mathbb{E}_{z_0 = E(x), \epsilon, t} \left[ \|\epsilon - F_\theta(z_t, t)\|^2 \right], \tag{19}$$

where $z_t = \sqrt{\alpha_t} z_0 + \sqrt{1 - \alpha_t} \epsilon$ and $\alpha \in (0, 1)$ is the cumulative product of noise factors up to timestep $t$.

However, the training objective $\mathcal{L}_{\text{denoise}}$ is decoupled from the final data-space generation goal:

$$\mathcal{L}_{\text{data}} = \|R(F_\theta(z_T)) - x\|^2. \tag{20}$$

Since the renderer $R$ is fixed and not involved in backpropagation during the diffusion training, the latent diffusion model is not explicitly encouraged to produce latents that yield faithful reconstructions after rendering. This mismatch can lead to error accumulation and unpredictable degradation in the final output quality.

## B  DETAILED EXPERIMENT SETTINGS

### B.1  CONSISTENCY TRAINING SCHEDULE

Following Karras et al. (2022); Song et al. (2023), we define $c_{\text{skip}}(t) = \frac{\sigma_{\text{data}}^2}{t^2 + \sigma_{\text{data}}^2}$ and $c_{\text{out}}(t) = \frac{t \cdot \sigma_{\text{data}}}{\sqrt{t^2 + \sigma_{\text{data}}^2}}$, where $\sigma_{\text{data}}^2$ represents the variance of the (normalized) data. We set $\sigma_{\text{data}}^2 = 0.5$ for all datasets. We follow ECT Geng et al. (2024) to use a continuous-time training schedule for consistency training. Specifically, we utilize overlapping intervals in consistency models, enabling the factorization $p(t, r) = p(t)p(r|t)$ and continuous sampling of infinite $t$ values from a noise distribution $p(t)$, i.e., LogNormal($P_{\text{mean}}, P_{\text{std}}$), with $P_{\text{mean}} = -1.1$, $P_{\text{std}} = 2.0$ and $r \sim p(r|t)$. The mapping function $p(r|t)$ adjusts over training iterations to reduce $\Delta t = (t - r) \rightarrow dt$. We parameterize $p(r|t, \text{iters})$ as

$$\frac{r}{t} = 1 - \frac{1}{q^a} n(t) = 1 - \frac{1}{q^{\lfloor \text{iters}/d \rfloor}} n(t),$$

where $n(t) = 1 + k\sigma(-bt)$, $\sigma(\cdot)$ is the sigmoid function, "iters" represents training steps and $d$ is a hyperparameter controlling how quickly $\Delta t \rightarrow dt$. By default, we set $q = 2$, $k = 8$, and $b = 1$. To ensure $r \geq 0$, $r$ is clamped appropriately. Initially, $r/t = 0$, which aligns with diffusion pretraining. To simplifying the models, we use $L_2$ norm as the distance metric $d(\cdot, \cdot)$. We follow CMs Song et al. (2023) and ECT Geng et al. (2024) to set $t_{min} = 0.003$ and $t_{max} = T = 80$. The reconstruction loss time threshold $t_{\epsilon_1}$ is then set to 0.007, which is about $2 \times t_{min}$, and the diffusion loss time threshold $t_{\epsilon_2}$ is set to 0.0865 so that diffusion loss is applied to the first 25% interval.

### B.2  DETAILED NETWORK ARCHITECTURES

In this part, we introduce the detailed architecture of our hyper-network generator $G_\theta$. The full architecture of $G_\theta$ is presented in Figure 5. Specifically, $G_\theta$ contains signal encoder blocks to extract features from the support observation set conditioned with noise level $t$, INR decoder blocks to predict the INR Tokens (the representation vector for the signals), and the INR functions that generate the query observation set with given INR Tokens and query transformations.

**Signal encoder blocks.** These blocks are responsible for extracting features from noisy images conditioned with the noise level and producing data tokens that guide the generation of the image

---

**Algorithm 1** Implicit Neural Representation Consistency Training (INRCT)

---

**Input:** Dataset $\mathcal{I}$, Generator $G_\theta$ with parameter $\theta$, INR function $M$, noise distribution $p(t, \text{Iters})$, mapping function $p(r \mid t, \text{Iters})$, weighting function $w(\cdot)$, EMA decay rate schedule $\mu(\cdot)$, reconstruction loss threshold $t_{\epsilon_1}$, diffusion loss threshold $t_{\epsilon_2}$, lr $\eta$, weights $\lambda_1, \lambda_2, \lambda_3$.

**Init:** $\theta^- \leftarrow \theta$, $\text{Iters} = 0$.

**repeat**

    Sample $O \sim \mathcal{I}$, $O^s \sim O$, $O^q \sim O$, $\mathbf{z} \sim p(\mathbf{z})$, $t \sim p(t)$, $r \sim p(r \mid t, \text{Iters})$

    Compute $O_t^s = O^s + t \cdot \mathbf{z}, O_r^s = O^s + r \cdot \mathbf{z}, O_t^q = O^q + t \cdot \mathbf{z}, O_r^q = O^q + r \cdot \mathbf{z}, \Delta t = t - r$

    $\mathcal{L}_{\text{gene}} \leftarrow d\left(f_{\boldsymbol{\theta}}\left(O_t^s, O_t^q, t\right), f_{\boldsymbol{\theta}^-}\left(O_r^s, O_r^q, r\right)\right),$

    $\mathcal{L}_{\text{recon}} \leftarrow d\left(T_q(M_{G_\theta(O_t^s, t \mid t < t_{\epsilon_1})}), O_q\right)$

    $\mathcal{L}_{\text{diff}} \leftarrow d\left(f_\theta(O_s^t, O_q, t < t_{\epsilon_2}), O_q\right)$

    $\mathcal{L}_{\text{INRCT}} = \omega(t)\left(\lambda_1 \mathcal{L}_{\text{gene}} + \lambda_2 \mathcal{L}_{\text{recon}} + \lambda_3 \mathcal{L}_{\text{diff}}\right)$

    $\theta \leftarrow \theta - \eta \nabla_\theta L, \theta^- \leftarrow \text{stopgrad}\left(\mu(\text{Iters})\theta^- + (1 - \mu(\text{Iters}))\theta\right), \text{Iters} \leftarrow \text{Iters} + 1$

**until** convergence **return** $\theta$

---

function for denoised images. We primarily follow the DiT framework Peebles & Xie (2023) and employ adaLN-Zero Transformer blocks to build the encoder blocks for feature extraction. As depicted in Figure 5 (b), we use a linear layer to regress the dimension-wise scale and shift parameters $\alpha$, $\gamma$, and $\tau$ from noise level embeddings.

**INR decoder blocks.** These blocks generate the INR tokens $\beta$ for the INR function based on the features extracted by the signal encoder blocks. Prior to training, we randomly initialize the learnable INR tokens according to the shape of the required INR tokens $\beta$. As shown in Figure 5 (c), we design an INR decoder block that uses multi-head cross attention to merge the data feature with INR tokens and predict the INR tokens for the INR function corresponding to the denoised images. We mainly follow ANR Zhang et al. (2024) to design the architecture for INR decoder.

**INR functions.** As shown in Figure 5 (d), we choose ANR Zhang et al. (2024) as the INR functions for image modality as it has presented superior representation ability for image, and the basic MLP Chen & Wang (2022) as the INR functions for 3D NeRF modality as it is more stable than ANR on 3D NeRF. After obtaining the INR Tokens $\beta$ from the INR decoder blocks, we render the images or query projections differentially to ensure the entire pipeline is differentiable.

For image modality, we define the INR function as an ANR with instance-agnostic parameters optimized during the training. This ANR-based INR function accepts the instance-specific INR Tokens and the queried coordinates and output the RGB corresponding to the queried coordinates. Given a resolution of $H \times W$, we follow the approach in Zhang et al. (2024) to sample a variational coordinate list $\{(\frac{i}{H}, \frac{j}{W})\}_{i,j}$ with small deviations, where $i \in [0, H)$ and $j \in [0, W)$. We then query the RGB value at each coordinate, which can be reshaped to form the complete image. Thanks to the variational coordinates, our pipeline supports realistic any-resolution image generation without artifacts.

For the 3D NeRF modality, we define the INR functions as an MLP with all parameters are generated from the INR Tokens. The queried 3D coordinates are sampled given the query transformation and the MLP would output the RGB and the density $\sigma$ at the corresponding coordinates. A differential volume rendering is applied based on the projection rays to obtain the projections from the query transformation.

To enable the hyper-network generator with sufficient capacity to generate realistic objects, we set the number of signal encoder blocks N = 8 and the number of INR decoder blocks M = 6. For both signal encoder blocks and INR decoder blocks, the attention dimensions are set to 768 and the feedforward dimensions are set to 3072. We adopt ANR Zhang et al. (2024) with a Localized Attention Layer with 512 hidden dim and 256 token length, followed by a 5-layer MLP with 512 width and ReLU activation for image modality. We adopt MLP with a depth of 6 and width of 256, with ReLU activation and positional embedding for the 3D NeRF modality.

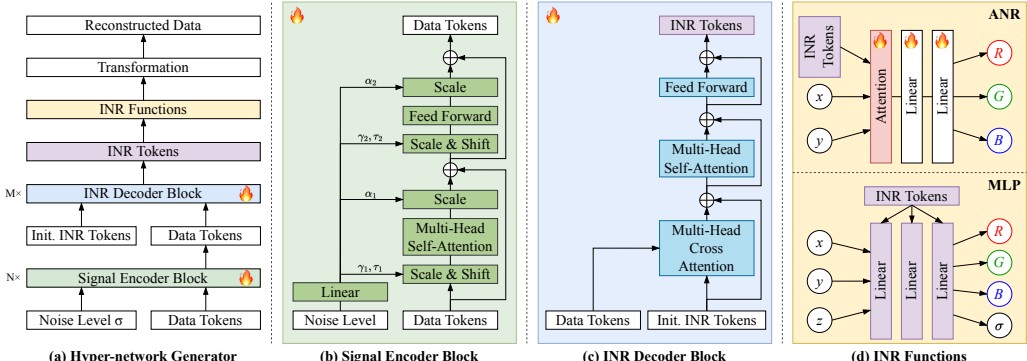

Figure 5: (a) The detailed architecture of our proposed hyper-network generator. The learnable parameters are in the signal encoder blocks and INR decoder blocks. (b)&(c) The detailed implementation of the signal encoder block and INR decoder block. (d) The INR functions to represent a continuous image (ANR Zhang et al. (2024)) and 3D NeRF object (MLP Chen & Wang (2022)).

### B.3 OPTIMIZATION PROCEDURES

We follow Peebles & Xie (2023) and Chen & Wang (2022) to split the $32 \times 32$ and $64 \times 64$ images with patch size 4, and the $128 \times 128$ images with patch size 8. All models are optimized with a batch size of 128 on 4 V100 GPUs with 32 GB. For the efficient training scheduler, the weight of the three loss terms is set to $\lambda_1 = 0, \lambda_2 = 1, \lambda_3 = 0$ to focus on the reconstruction task in the early training phase, and is set to $\lambda_1 = 1, \lambda_2 = 1, \lambda_3 = 1$ for the full training. Our experiments demonstrate that the three loss terms maintain comparable magnitudes, and the framework exhibits robustness to the values of $\lambda$, which indicates that varying $\lambda$ coefficients by an order of magnitude (e.g., setting to 0.1 or 10) shows minimal impact on model performance.

We optimize all models using the Rectified Adam optimizer with the learning rate=$1e-4$ and early stopping to prevent divergence when $\Delta t$ becomes too small. The complete training process employs an efficient scheduler that first focuses exclusively on the reconstruction objective before transitioning to full training. Specifically, CIFAR-10 undergoes 1,725 total epochs (initial 125 epochs for reconstruction), CelebA-HQ completes 2,125 epochs (first 125 epochs for reconstruction), and SRN Cars trains for 6,000 epochs (initial 1,000 epochs for reconstruction). The total training times are 48, 55, and 74 hours for CIFAR-10, CelebA-HQ, and SRN Cars, respectively. Note that we do not apply the EMA strategy on the initial training for reconstruction because we find that the training for INR encoding is quite stable and EMA would slow down the convergence of the optimization process. Our implementation is based on PyTorch with CUDA 11.8.

## C ADDITIONAL EXPERIMENT DETAILS AND RESULTS

### C.1 IMAGES

We use the CIFAR-10 dataset Krizhevsky et al. (2009) and the CelebA-HQ dataset Karras (2017) for our work. For the CIFAR-10 dataset, we use the original train-test split categories, i.e., 50000 images for the training set and 10000 images for the testing set. For the CelebA-HQ dataset, we follow Functa Dupont et al. (2022a) and mNIF You et al. (2023) to divide the entire dataset into 27,000 images for training and 3,000 images for the test split. To measure the quality of generated images, we compute Fréchet inception distance (FID) score Heusel et al. (2017), precision, recall, and F1 scores Kynkäänniemi et al. (2019) between sampled images and images in a train split. For evaluation of the INR encoding, we use the PSNR to measure image similarity between reconstructed data and the ground truth data in the testing set.

We present more visualization performance of CIFAR-10 dataset and CelebA-HQ dataset in Figure 6 and Figure 7. The results show that INRCT can generate realistic and diverse results that closely match the distribution of the training data.

Table 6: Time efficiency to generate 50,000 samples.

|  | NFE | Time to generate 50,000 samples ↓ |
|---|---|---|
| mNIF You et al. (2023) | 1000 | 90 mins |
| **INRCT(Ours)** | **1** | **5 mins** |
| **INRCT(Ours)** | **2** | **10 mins** |

Table 7: Training cost analysis of INRCT with the baseline methods in CelebA-HQ $64^2$ dataset.

|  | Training Parameter (M) | | | Training images (kilo imgs) | | | Batch_size |
|---|---|---|---|---|---|---|---|
|  | Encoding | Generation | Total | Encoding | Generation | Total | per GPU |
| Functa | 3.3 | 158 | 161.3 | 128,000 | 128,000 | 256,000 | 1 |
| mNIF | 4.6 | 276 | 280.6 | 21,600 | 27,000 | 48,600 | 8 |
| INRCT | - | - | 145 | - | - | 57,375 | 32 |

We also show the time efficiency to generate 50,000 samples for the CelebA-HQ dataset with a single 48 GB A6000 GPU in Table 6. We find that INRCT requires about 10 minutes to generate 50,000 two-step inference results and about 5 minutes for 50,000 one-step inference results. However, mNIF relying on DDPM requires 1000 steps to generate samples, taking about 90 minutes to generate 50,000 samples. Our INRCT generates better samples in about one-tenth of the time.

## C.2 NeRF Objects

Following Functa Dupont et al. (2022a) and mNIF You et al. (2023), we adopt the SRN Cars dataset Sitzmann et al. (2019) for NeRF modeling. The SRN cars dataset is divided into training, validation, and test sets. The training set contains 2,458 scenes, each with 128×128 resolution images captured from 50 random views. The test set consists of 704 scenes, with 128×128 resolution images taken from 251 fixed views in the upper hemisphere. During training, we randomly choose 1 view as the support observation set and 1 view as the query observation set. Therefore, our generator would predict the NeRF given only 1 view and is supervised with another view, acting as a single-image-to-3D process. For evaluation, we follow the NeRF scene setting used in Functa Dupont et al. (2022a) and mNIF You et al. (2023), which employs the 251 fixed views from the test split. As a result, we can calculate the FID score by comparing rendered images with those in the test set, ensuring the view statistics are consistent. To evaluate INR encoding, we still use PSNR to measure image similarity between reconstructed NeRF views and the ground truth views in testing set.

In Figure 8, we show the full 251 views of 2 examples generated by INRCT. We observe that INRCT can generate consistent images across all views from all different projection poses. In Figure 9, we present all generated samples from INRCT with random projection views and the ground truth from SRN cars datasets. Results show that INRCT can generate realistic and diverse NeRF objects.

## C.3 More Comparison to the Baseline Methods

We present the total training parameters and the training cost of INRCT with two existing baseline methods in Table 7. We find that INRCT has fewer parameters (145M) than both Functa (161M) Dupont et al. (2022a) and mNIF (281M) You et al. (2023), which construct the denoiser with U-Net-based generators. This result indicates that the performance improvements in both encoding and generation tasks come from the end-to-end training framework rather than increased model capacity.

Due to variations in training mechanisms, hardware configurations, and batch sizes across different methods, we follow ECT Geng et al. (2024) to use the total number of images processed (calculated as batch size × training iterations, measured in kilo imgs) as a quantitative metric for training cost analysis. The results demonstrate that INRCT achieves better model performance than mNIF You et al. (2023) while using a similar total number of training images, which is significantly fewer than required by Functa Dupont et al. (2022a).

For training hardware requirements, the meta-learning-based encoding process of Functa Dupont et al. (2022a) is highly memory and computation-intensive, requiring first-order gradient computa-

Table 8: Comparison of models on CelebA-HQ dataset.

| Models | # Params (M) ↓ | NFE ↓ | Latency (s) ↓ | FID on CelebA-HQ ↓ |
|---|---|---|---|---|
| Non Diffusion-Based Generative Methods | | | | |
| GEM | - | - | - | 30.4 |
| GASP | 34.23 | 1 | 0.0109 | 13.5 |
| CIPS | - | 1 | - | 15.4 |
| Diffusion-Based Generative Methods | | | | |
| DPF | 62.4 | 1000 | - | 13.2 |
| Functa | 161.3 | 1000 | - | 40.4 |
| mNIF(S) | 280.6 | 1000 | 0.108 | 21.0 |
| DDMI | 2335.4 | 1000 | 0.822 | 9.74 |
| **INRCT (1-step)** | **145.5** | **1** | **0.006** | **12.7** |
| **INRCT (2-step)** | **145.5** | **2** | **0.012** | **7.88** |

tion and limiting its batch size to just 1 on a 48GB GPU. While the auto-decoding approach from mNIF You et al. (2023) shows better efficiency with batch_size=8 on a 48GB GPU, INRCT shows best memory efficiency by supporting batch_size=32 on a 32GB GPU.

These comparisons clearly show that INRCT matches or even surpasses the baseline methods in terms of total training parameters, total training cost, and GPU memory efficiency, making it a more practical and scalable solution for INR-based generation tasks.

In addition, we extend Table 2 to include more efficiency metrics, including the learnable parameters and the generation latency. The latency is defined as the average time (in seconds) required to generate a single image, measured over 1,000 samples. As shown in Table 8, INRCT demonstrates a clear performance advantage over previous non-diffusion-based generative methods, while also achieving improvements in both performance and efficiency compared to diffusion-based approaches.

## D LIMITATIONS AND FUTURE WORKS

In this section, we discuss the limitations and future works of the current INRCT framework. One limitation of our work is that we have only tested on relatively simple data, such as low-resolution images and simple NeRF objects, due to hardware restriction. We will explore the challenges and potential solutions for scaling this method to more complex signals.

First, for larger resolution images, the existing consistency training in the data space faces complexity issues. Directly applying consistency training on the higher-dimensional noise space requires larger network capacities and more computational resources, including more GPU memory and longer training times Geng et al. (2024); Song & Dhariwal (2023). Current consistency training methods Luo et al. (2023) often follow the approach of latent diffusion models, where a VAE is used to reduce the dimensionality of images, and then consistency training is performed in the latent space. This approach contradicts the primary motivation of our work, which aims to train directly in the data space to provide more precise supervision signals. In fact, INRCT can avoid training in the latent space. By introducing the INR intermediate, INRCT decouples the input and output of the consistency function and supports any-resolution image sampling. Specifically, the input $O^s$ to the network and the output $O^q$ as the supervision signal can now be in different resolutions. $O^q$ can be a high-resolution image, providing a higher-quality supervision signal, while $O^s$ can be a lower-resolution image, making consistency training and model processing more efficient. By employing this strategy, it is possible to efficiently scale INRCT to high-quality, high-resolution image generation.

Then, for complex NeRF generation, the INR function in the current INRCT lacks sufficient representation and generalization capability for complex NeRF signals. This is primarily because we have only used a simple MLP-based INR function to represent NeRF and employed basic volume rendering techniques to obtain projections from different angles. In the future, we will explore more advanced NeRF techniques, such as tri-plane representations Hong et al. (2024) and anti-aliasing techniques by Mip-NeRF Barron et al. (2021), which can enhance the performance of INRCT for NeRF signals.

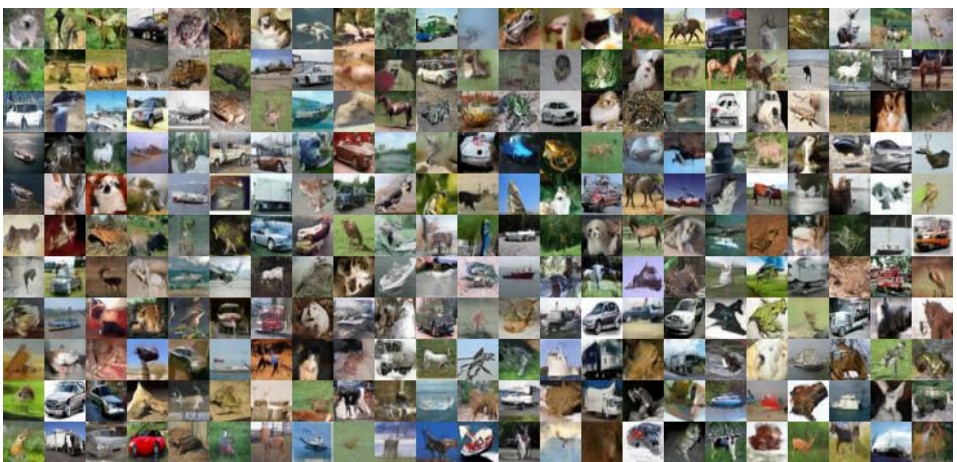

Generated Samples By INRCT

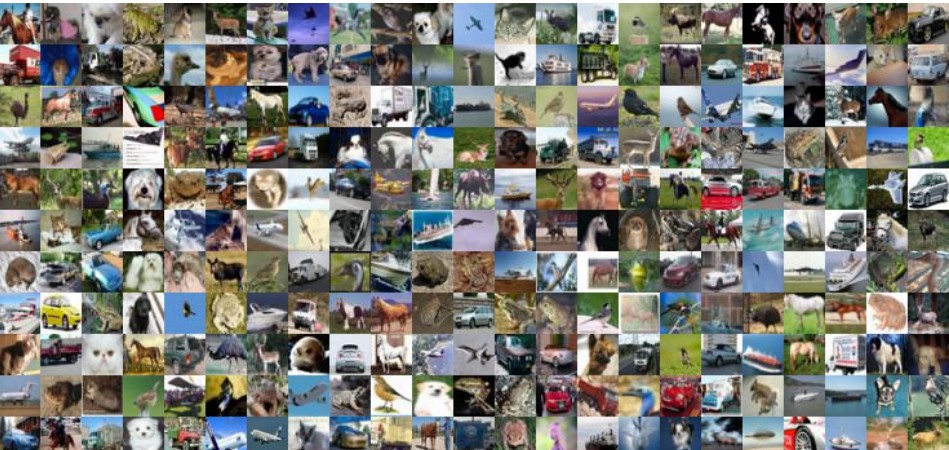

Ground Truth in CIFAR-10 Dataset

Figure 6: Generated samples from our INRCT trained on the CIFAR-10 dataset and the training samples from the CIFAR-10 dataset. We can observe that the generated samples by INRCT are very close to the training ground truth.

In conclusion, INRCT provides a more efficient, accurate, and unified training and inference framework for INR-based encoding and generation models. We believe there are still many aspects that can be further explored, and it has significant potential to drive further advancements in the field of generative models.

## E  BROADER IMPACTS

Generation is a widely discussed problem. Deep learning generative models can greatly impact society. Positively, they can enhance creative industries, healthcare, and education by generating art, improving medical imaging, and personalizing learning. However, they also pose risks, such as creating deepfakes, spreading misinformation, and raising ethical concerns about data privacy and bias. Balancing these benefits and risks is essential for their responsible use.

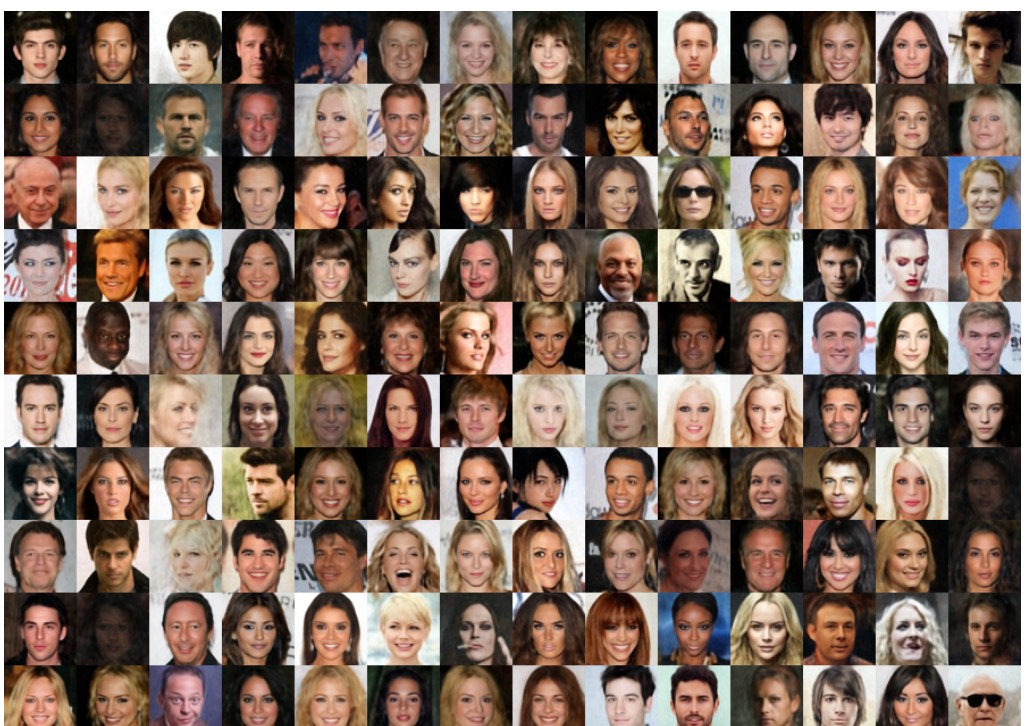

Generated Samples from mNIF

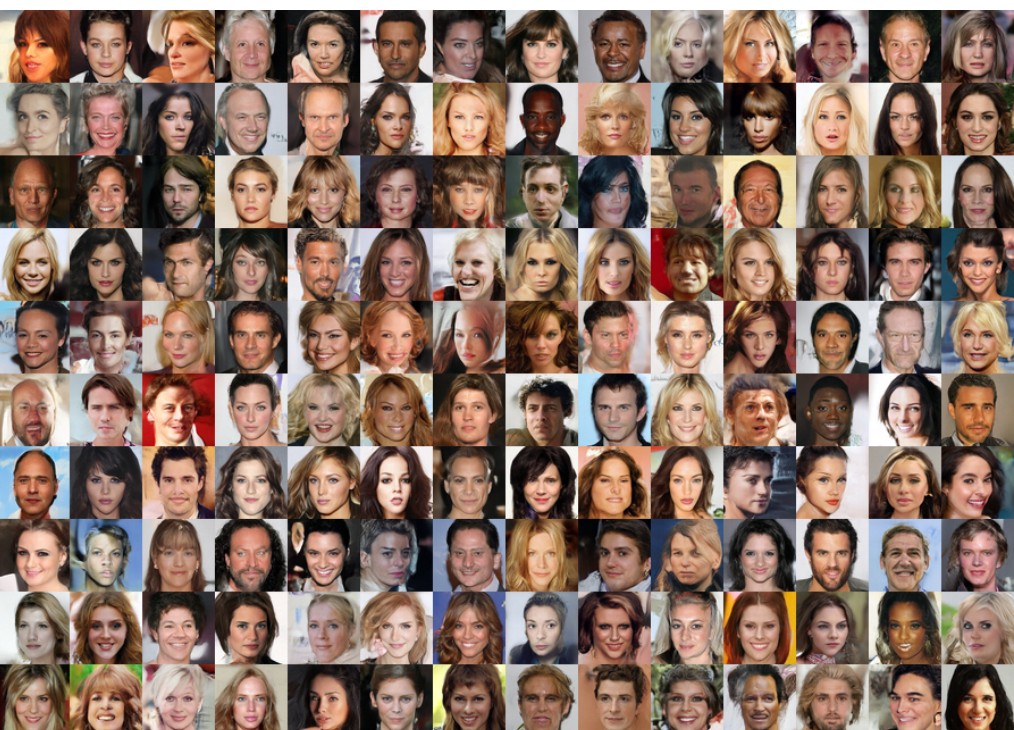

Generated Samples from INRCT

Figure 7: The generated samples from mNIF (top) and INRCT (bottom) on the CelebA-HQ dataset. We find that mNIF sometimes generates repeated images (see the images at index [1,1],[9,1] and [7,14]). In contrast, INRCT can generate realistic and diverse samples.

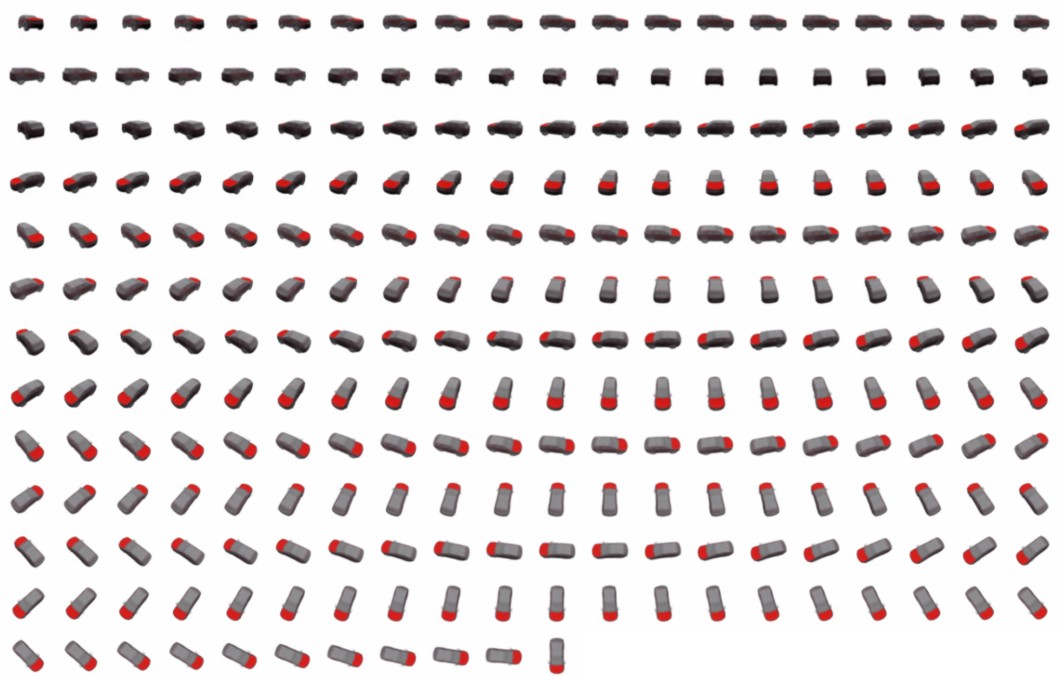

Full View of the Generated Example 1

Full View of the Generated Example 2

Figure 8: The full 251 views of 2 examples generated by INRCT. The results show that INRCT can generate consistent images across all views from different projection poses.

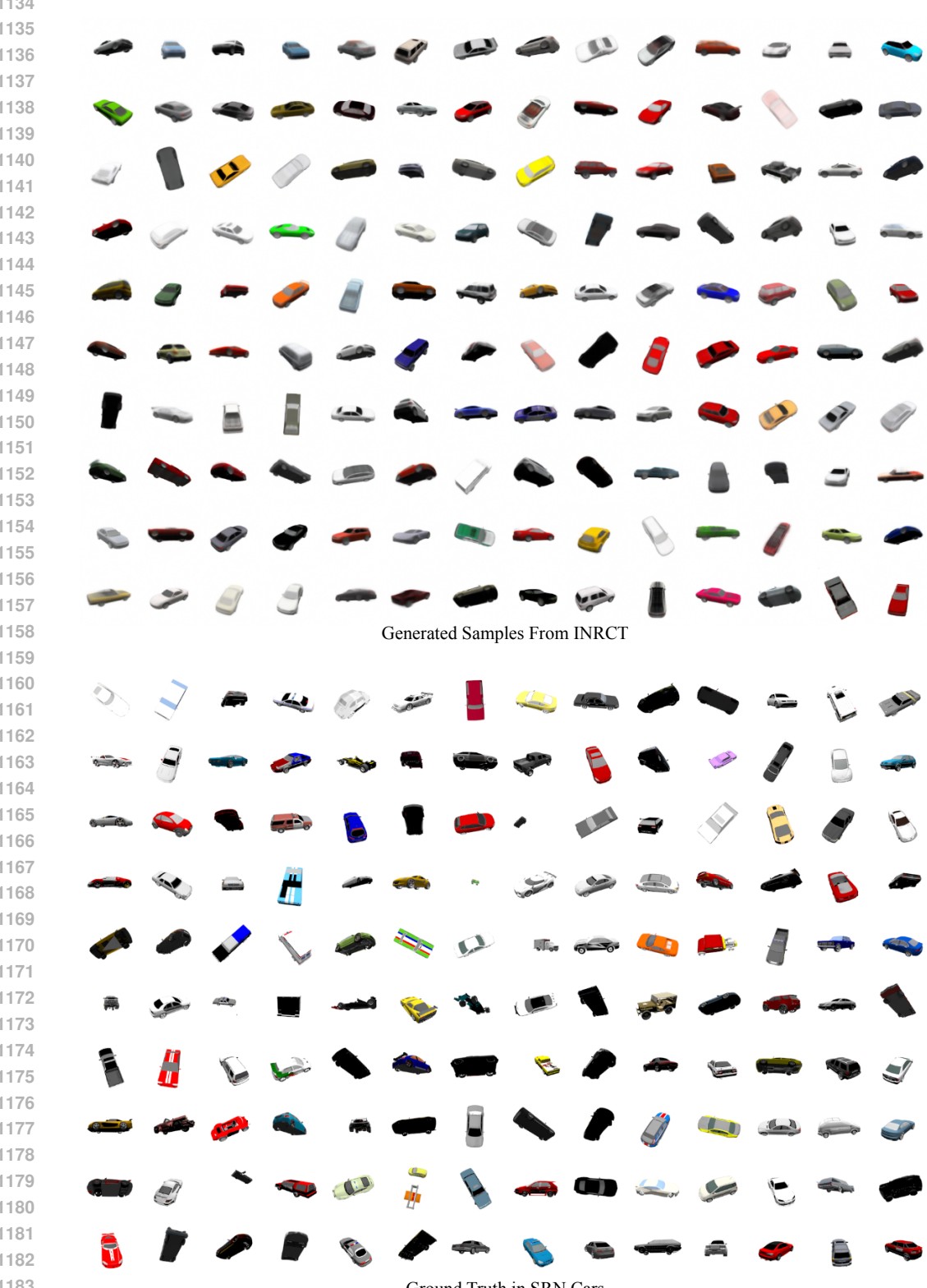

Generated Samples From INRCT

Ground Truth in SRN Cars

Figure 9: The generated samples from INRCT (top) and the ground truth (bottom) from SRN cars datasets. We generate NeRF objects with random views. The results show that INRCT can generate realistic and diverse NeRF objects.

