# OpenReview forum: "INRCT: An End-to-End Framework for Encoding and Generating Implicit Neural Representation"
_ICLR.cc/2026/Conference — ICLR 2026 Conference Withdrawn Submission_

### Official Review · Reviewer_wcDN · 2025-10-22

**Soundness:** 3
**Presentation:** 3
**Contribution:** 3
**Rating:** 6
**Confidence:** 2

**Summary:**

This paper presents INRCT, an end-to-end, modality-agnostic generative framework that unifies implicit neural representation (INR) encoding and generation using a single model via consistency training. The core innovation is to replace the standard two-stage latent diffusion pipeline with a hyper-network trained as a denoiser, performing consistency-based few-step diffusion directly in data space and outputting INR parameters. This single model supports both INR encoding (signal-to-INR) and INR generation (noise-to-INR), with a cross-modal loss that combines generation and reconstruction objectives. Experiments on standard image and 3D generative modeling benchmarks demonstrate improvement over prior two-stage diffusion-based INR methods in both generation quality, reconstruction fidelity, and efficiency, and show qualitative flexibility including multi-resolution and single-image-to-3D generation.

**Strengths:**

1. Instead of using two separate models like previous works, INRCT achieves both INR encoding and generation within a single, end-to-end trainable hyper-network, simplifying the overall architecture and reducing compounded error from decoupled stages.

2. The method is versatile—demonstrating applicability across images and 3D NeRF scenes. The cross-modal training objective extends consistency training to multiple domains.

3. Experiment results (Table 1, Table 2, and Table 3) show that for both image (CIFAR-10, CelebA-HQ) and 3D object (SRN Cars) tasks, INRCT matches or exceeds the FID, IS, PSNR, and other relevant metrics of two-stage approaches, while also being faster (Table 6, Table 8). Ablation analysis (Table 5) substantiates the contribution of each component.

**Weaknesses:**

1. While the proposal to unify encoding and generation is well-motivated, most core technical elements (such as probability flow ODE, consistency loss, hyper-networks for INR generation) are adaptations of previous works (Song et al., 2023; Geng et al., 2024; Dupont et al., 2022a; You et al., 2023; Park et al., 2024). The direct application of these ideas is potentially incremental and should be somewhat more deeply contrasted in the method.

2. As acknowledged in Section D, experiments are only performed on low-resolution (CIFAR-10, $32\times32$, CelebA-HQ $64\times64$) and fairly simple NeRF scenes. The method’s claimed scalability and flexibility to more complex generative tasks remains speculative.

3. Although the method is proposed as scalable (L938), the authors themselves note that training in high-dimensional data space is still a substantial challenge. A more concrete discussion (or at least an experiment with larger-scale images or more complex NeRFs) would be welcome.

**Questions:**

1. Can the authors elaborate, with more quantitative estimates or empirical evidence, on the memory and runtime consequences of applying INRCT to higher-resolution or more complex domains? Could hybrid latent-data approaches (as hinted in Section D) be unified with INRCT?

2. Why is the combination $\mathcal{L}{\text {gene }}$, $\mathcal{L}{\text {recon }}$, and $\mathcal{L}_{\text {diff }}$ superior to standard consistency loss or other combinations? Are there settings where the diffusion loss, in particular, might hinder convergence?

---

### Official Review · Reviewer_Cpyd · 2025-10-28

**Soundness:** 3
**Presentation:** 3
**Contribution:** 2
**Rating:** 2
**Confidence:** 4

**Summary:**

- This paper introduces INRCT (Implicit Neural Representation Consistency Training), a novel end-to-end framework for encoding and generating signals using Implicit Neural Representations (INRs).
- The main goal of the method is to unify INR encoding and generation into a single end-to-end trainable model, avoiding the limitations of existing two-stage approaches that rely on separate encoding and diffusion models.
- INRCT uses a hyper-network to directly map noisy observations to the INR parameters of the clean signal.
- It leverages consistency training from diffusion models to enable few-step generation.
- The model is trained with a combined objective:
  - Generation loss: Enforces consistency between any two adjacent noisy points in the PF-ODE.
  - Reconstruction loss: Enables accurate INR encoding from clean signals at the boundary time-steps.
  - Diffusion loss at smaller time-steps for stabilizing the training.
- Evaluated on CIFAR-10, CelebA-HQ, and SRN Cars datasets for image and 3D NeRF generation.

**Strengths:**

- The paper is well written.
- First end-to-end framework for INR encoding and generation.
- Modality-agnostic: applicable to 2D images and 3D NeRFs.
- Supports any-resolution image generation and single-image-to-3D NeRF synthesis.
- Outperforms existing INR-based generative models (e.g., Functa, mNIF, DDMI) in both generation quality (FID, IS) and efficiency (NFE, latency).
- End-to-End Training: Eliminates error accumulation from separate encoding and diffusion stages.
- Efficiency: Few-step (even one-step) generation reduces inference time significantly.
- Achieves better or competitive FID, IS, and PSNR scores across datasets.
- Combines multiple losses and a phased training scheduler for stable convergence.
- Ablation studies examining each objective term is provided.

**Weaknesses:**

- Scalability: Tested only on low-resolution images and simple 3D scenes.
- Still lags behind some discrete image generation methods (e.g., CT) in FID on CIFAR-10.
- The paper claims generalization to any resolution. Yet when examining the results in Figure 4b, it is obvious that the method does not generalize well beyond the training resolution (64\$\times\$64).
- Mapping between data space and INR space introduces optimization challenges that does not worth the effort.
- The paper does not present any evaluation of method on \$NFE>2\$.

**Questions:**

- As far as I can tell there is no proper definition of \$\\{c_i\\}\$ (the input of the INR function) in the paper.
- How does the method perform when examined on scenarios where \$NFE>2\$?

---

### Official Review · Reviewer_9wZ9 · 2025-11-02

**Soundness:** 3
**Presentation:** 3
**Contribution:** 3
**Rating:** 6
**Confidence:** 3

**Summary:**

The paper introduces INRACT, an end-to-end generative framework using Implicit Neural Representations (INRs). INRACT operates diffusion on raw data and leverages consistency training to achieve fast inference. Given a source and a target observation set, the framework first encodes noisy source data into INR representation, then uses the INR function of the target to decode it back to raw data domain. This whole stack acts as the denoiser and subjects to consistency training. In addition, reconstruction is added as an auxiliary loss to stablize training and achieve better result. The paper shows improved performance compared to other INR based methods, although the performance still falls short compared to non-INR based diffusion.

**Strengths:**

The paper explores a very different approach for generative modelling which uses INR and diffusion in the raw data domain.

**Weaknesses:**

Scalability seems to be an issue. It's unclear whether this approach can scale to larger models or high resolution data.

The paper doesn't convincingly show the benefits of operating through INR representation.

**Questions:**

1. For diffusion loss at small time-steps (line 300), is the implementation equivalent to extending the reconstruction loss for $T_q(M_{G_θ(O^s_t,t)}) \quad \forall t < t_{\epsilon}$ ?

2. There is inconsistency in using $s$ and $q$ as subscript or superscript. E.g. eq 12 use q as subscript while line 301 uses as superscript.

3. Since the consistency model parameterization (eq 3) doesn't fully satisfy the need of INRACT (the need of adding reconstruction loss), could the authors comment on whether there are other parameterizations that might work out better for INRACT?

4. What is the concrete form of $w(t)$ used in training?

5. Could the authors provide more discussion about CelebA-HQ samples from INRACT? While authors argue that the samples are more "realistic and diverse" (line 421), it feels to me that they are less coherent globally and often look less natural.

6. In table 5, it seems that adding reconstruction loss alone at $t=0$ has a worse performance than not adding it at all. Could the authors comment on this?

7. Could the authors discuss more on the issue of scalability? It seems to me that using latent or lower resolution as source would defeat the purpose of diffusion on raw data, as both result in loss of information.

8. Could the authors provide some experimental results on sampling speed and high resolution generation performance compared against diffusion or consistency model? The authors advocate that one of INR's benefits is the flexibility of generating samples at larger resolution, but how does it perform compared to interpolation a diffusion/consistency model?

---

### Note · Authors · 2025-11-21

**Comment:**

We sincerely appreciate the valuable feedback provided by the reviewers. We are happy that the paper received two positive and encouraging assessments. However, we were surprised by one reviewer’s significantly lower overall rating, which appeared inconsistent with the content of the written comments. To further strengthen the manuscript, we decide to withdraw the submission.

**Withdrawal Confirmation:**

I have read and agree with the venue's withdrawal policy on behalf of myself and my co-authors.